# Effect of Hydroxymethylfurfural and Low-Molecular-Weight Chitosan on Formation of Acrylamide and Hydroxymethylfurfural during Maillard Reaction in Glucose and Asparagine Model Systems

**DOI:** 10.3390/polym13121901

**Published:** 2021-06-08

**Authors:** Hong-Ting Victor Lin, Der-Sheng Chan, Ling-Yu Kao, Wen-Chieh Sung

**Affiliations:** 1Department of Food Science, National Taiwan Ocean University, Keelung 202301, Taiwan; HL358@mail.ntou.edu.tw (H.-T.V.L.); rachael20148888@yahoo.com.tw (L.-Y.K.); 2Department of Information Technology, Lee-Ming Institute of Technology, New Taipei City 243083, Taiwan; dschan@ms58.hinet.net

**Keywords:** acrylamide, low-molecular-weight chitosan, 5-hydroxymethyl furfural, kinematic viscosity, Maillard reaction products

## Abstract

The aim of this research was to investigate the effects of the addition of 0.5% hydroxymethylfurfural (HMF) and low molecular chitosan on acrylamide and HMF formation in a food model system, which contains 0.5% glucose, asparagine, and HMF within 30 min of heating at 180 °C. At an interval of 10 min, all solutions were evaluated in the following aspects: reducing sugar, asparagine, acrylamide, HMF content, pH, Maillard reaction products, kinematic viscosity, and color. After heating for 10 min, the kinematic viscosity of solutions containing chitosan reduced significantly. The values of the acrylamide, HMF, and absorbance increased at OD_294_ and OD_420_ (optical density measured at 294 nm and 420 nm) of solutions. Experimental results showed that low-molecular-weight chitosan might be hydrolyzed into much lower molecular weight, followed by the decrease in kinematic viscosity of the solution at pH lower than 6 and the increase in the formation of acrylamide after heating for 30 min.

## 1. Introduction

Acrylamide and 5-hydroxymethylfurfural (HMF) are undesirable carcinogens and are found as by-products of Maillard reaction in heat-processed foods containing starch [1], such as bread, cereals, and oils, used in the preparation of foods, including potato chips and French fries [2]. Maillard reaction among reducing sugars, such as fructose, glucose, and asparagine, is the major mechanistic pathway in the formation of food contaminants [3,4]. The major route of the formation of acrylamide involves a reactive carbonyl and asparagine through intermediates, which include a decarboxylated base, 3-aminopropioamide, and a Schiff base. The intermediate Schiff base eliminates either ammonia or substituted imine and decarboxylizes to form acrylamide during heating [5]. Asparagine is one of the most important amino acids that will suitably react with dicarbonyl compounds, which becomes a precursor in the production of acrylamide as Strecker degradation in the Maillard reaction [5]. The initial step in this process is the Schiff base formation, followed by addition of nucleophilic asparagine to the partially positive carbonyl carbon of the dicarbonyl compound. The mechanism of the reaction, which involves the loss of a proton from nitrogen and the gaining of a proton by oxygen, was proposed by Mottram et al. [6]. Formation of ^15^N-labeled acrylamide can be done by rearrangement of glucose and the nitrogen-15(amido)-labeled asparagine in reactions of the heated food model [3]. The degradation product of the Amadori rearrangement of the Maillard reaction is HMF, which is also similar to acrylamide [7]. HMF is also a heterocyclic compound formed by heating asparagine- and carbohydrate-rich foods [4] and by caramelization of reducing sugars under acidic conditions, which is suspected to have genotoxic and mutagenic effects [1]. In addition, HMF is also a naturally occurring biomass-derived C6 carbohydrate-based and being accessible by the acid-catalyzed dehydration of hexoses [8].

Chitosan is a linear polysaccharide and partially an N-deacetylated derivative of chitin from the crustacean shell, such as shrimps or crabs [9]. Chitosan have excellent properties, which include low toxicity, high biocompatibility, biodegradability, biocompatibility, and capacity for adsorption, making them potentials for biomedical, food, and pharmaceutical applications [10]. It has been used as natural antifungal and antibacterial preservatives [11]. However, the thermal process of chitosan addition in bakery or beverage products might influence the formation of harmful compounds, such as acrylamide and HMF. Formation of acrylamide cannot be derived solely by heating chitosan, asparagine, glucose, or fructose at 180 °C for 30 min, which confirms the requirement for Strecker degradation and the dicarbonyl reactant [6,12,13]. Maillard reaction products can be produced by reactions between the carbonyl groups of fructose, glucose, and chitosan-containing amino groups [12,13]. The browning intensity of Maillard reaction products can be increased by low-molecular-weight chitosan (50–190 kDa), which simultaneously reduce the formation of acrylamide in 1% glucose–asparagine and fructose–asparagine food model systems [12,13]. Kinematic viscosity directly varies with the molecular weight of chitosan (50–190 kDa, 190–310 kDa, and 310–375 kDa); however, the viscosity was not significantly different among the chitosan solutions heated at 180 °C for 30 min [12]. Amino groups of chitosan can react with carbonyl groups of reducing sugars, ketones, or aldehydes to produce chitosan-sugar conjugates, which have been considered as Maillard reaction products [14]. Investigating the mitigation and formation of HMF and acrylamide during food model heating reaction and quantifying their effect on different ingredients could be very helpful in the food industry to develop a more practical strategy for reducing neo-formed contaminant content of processing food. In addition, the summary of mitigation options for the formation of acrylamide in raw materials and heat-processed foods, together with control or addition of other ingredients, were done by Codex [15]. In this work, 0.5% low-molecular-weight chitosan (50–190 kDa) were reacted with 0.5% glucose and 0.5% asparagine and the content of HMF and acrylamide were measured during 30 min of heating. The formation of HMF from caramelization of glucose or from glucose–asparagine and HMF–asparagine can be distinguished during heating at 180 °C for 30 min in the study. The content of acrylamide generated from HMF–asparagine and glucose–asparagine model systems was also investigated. This study infers that the two model systems of glucose–asparagine solutions were used to explore the impact of 0.5% of these substances on Maillard reaction. The possible effects of chitosan on formation of acrylamide and HMF generated from glucose and asparagine at different heating intervals was evaluated to understand these interactions. The effect of changes in Maillard intermediate compounds, brown pigments, HMF, acrylamide, pH, color, and kinematic viscosity were compared between aqueous control solution of glucose and asparagine, in addition to the test solutions with chitosan in 1% acetic acid.

## 2. Materials and Methods

### 2.1. Raw Materials and Chemicals

D(-)glucose, low-molecular-weight chitosan (50–190 kDa with deacetylation degrees >75%), ^13^C_3_-labeled acrylamide, and L-asparagine monohydrate were purchased from Sigma-Aldrich (St. Louis, MI, USA). 5-hydroxymethylfurfural was obtained from Acros Organics Company (Jersey City, NJ, USA). Acrylamide standard (99.9%) was procured from J.T. Baker (Phillipsburg, NJ, USA). All chemical reagents used in this study were of analytical grade. Oasis HLB (6 mL, 0.2 g) and Oasis MCX (3 mL, 0.06 g) solid phase extraction cartridges were procured from Waters (Milford, MA, USA).

### 2.2. Preparation of Maillard Reaction Products (MRPs)

MRPs were prepared by combining 0.5 g asparagine, 0.5 g chitosan, and 0.5 g glucose with 1% acetic acid following the method described by Chang et al. [16] with slight modifications. Solutions containing 0.5 g asparagine, 0.5 g glucose, and 0.5 g hydroxymethylfurfural in distilled water were also prepared. The pH value of each solution was first adjusted to 5.8 by adding 1 N sodium hydroxide (NaOH) and then 0.001 N NaOH to pH 6.0 and topped up with distilled water to 100 mL. Each solution was heated at 180 °C for 10, 20, and 30 min, followed by cooling in tap water.

### 2.3. Determination of Reducing Sugar, Asparagine Content, pH, MRPs, Acrylamide, Kinematic Viscosity, and HMF

Reducing sugars were measured by a dinitrosalicylic acid-reducing sugar assay described by Başkan et al. [17]. Calibration curve of the reducing sugar was prepared at 540 nm in the range of 0–6000 μg/mL using a Multi-detection reader (Synergy HT, BIOTEC instrument, Winooski, VT, USA). 

Asparagine in the solutions were extracted by the method of Bartolomeo and Maisano [18]. A portion (0.5 g) of each MRP solution was put into a 50-mL centrifuge tube and 20 mL of 0.1 N HCl was added. The centrifuge tube was placed in an ultrasonic water bath for 10 min and filled up to the mark at 25 mL with 0.1 N HCl. The solution was filtered with a filter paper. An aliquot (20 μL) of the filtrate was transferred into a glass vial and 100 μL of 0.4 M borate buffer (pH 10.2) was added and the mixture was vigorously shook. An aliquot (20 μL) of the mixture was transferred into a glass vial and 100 μL of 0.4 M borate buffer (pH 10.2) was added, followed by vortexing. The mixture was transferred into a glass vial and 20 μL of o-phthalaldehyde (OPA) was added, followed by vortexing for 60 s. Then, 20 μL of 9-fluoremenylmethyl chloroformate (FMOC-Cl) was added, followed by vortexing for 30 s. The mixture was diluted with 1280 μL of deionized distilled water and samples were subjected to derivatization. Chromatographic separation was performed at 40 °C and at a wavelength (λ) of 338 nm using Capcell Pak C_18_ AQ S5 column (5 μm, 4.6 mm × 250 mm) (Shiseido, Tokyo, Japan) following the method of Bartolomeo and Maisano [18]. Mobile phase A was 40 mM NaH_2_PO_4_ adjusted to pH 7.8 with NaOH, while mobile phase B raised the eluent B to 46% at a duration of 13 min. Afterward, washing in 100% B and equilibration in 0% B were done for a total analysis time of 20 min.

The pH of the MRP solution was measured with a pH meter (pH 510, Eutech Instruments Pte Ltd., Singapore) standardized with buffer solutions of pH 4.00 and 7.00 (AppliChem GmbH, Darmstadt, Germany).

Each solution was centrifuged at 21,900× *g* and 4 °C for 15 min and the supernatant was filtered through a 0.45-μm nylon filter and collected. UV-absorbing intermediates and browning intensity of the MRPs were measured at optical density measured at 294 nm and at 420 nm (OD_294_ and OD_420_), respectively, following the method of Ajandouz et al. [19]. An appropriate dilution was made using distilled water until the OD_294_ and OD_420_ were below 1 (Synergy HT Multi-detection reader, Biotek Instrument, Winooski, VT, USA). The acrylamide level from the Maillard reaction with internal standard was measured by high-performance liquid chromatography (HPLC) following the method of Chang et al. [12]. The acrylamide calibration curve was built in the range of 0–3125 ppb using a UV detector at 210 nm.

Kinematic viscosity was measured according to the time required for solutions to flow through the capillary tubes (Cannon-Fenske, No. 75 and 50, Cannon Instrument Company, State College, PA, USA). The tubes were equilibrated at 30 ± 0.2 °C for 10 min in a water bath (Tamson TMV-40, Zoetermeer, The Netherlands). The time required for a solution to be drained by gravity through the capillary tubes was recorded and this time was converted to a value for kinematic viscosity (cSt) using the extrapolated constant for the No. 50 and 75 capillary viscometers. To calculate kinematic viscosity, efflux time was multiplied by viscometer constant (cSt/s) [20].

HMF extraction from MRP solution was also determined by HPLC according to the method of Cai et al. [21] with some modifications. Each MRP solution was vortexed for 60 s and then placed in an ultrasonic water bath for 60 min and centrifuged at 27,216× *g* and 4 °C for 15 min. The HLB/MCX cartridge was conditioned with 5 and 3 mL of methanol followed by 5 and 3 mL of deionized distilled water. The supernatant was filtered with a 0.45-μm nylon filter and 3 mL of the filtrate was passed through an Oasis HLB/MCX cartridge to absorb the HMF before being discarded. The cartridge was washed with 0.5 mL of deionized distilled water and the filtrate was discarded. The cartridge was then washed with 3.0 mL of deionized distilled water and the eluate was collected in an amber glass tube and concentrated under vacuum (RV 10 digital, IKA, Staufen im Breisgau, Germany) at 40 °C for HPLC analysis.

The HPLC system (D2000) consisted of an L-2130 pump, L-2400 detector, L-2300 column oven, and L-2200 autosampler (Merck, Hitachi, Kent, UK). Chromatographic separation was done on a COSMOSIL 5C18-PAQ guard column (4.6 mm × 10 mm 3 ea/pkg) and COSMOSIL 5C 18-PAQ (5 μm, 4.6 mm × 250 mm) (Nacalai Tesque, Kyoto, Japan) using 25 °C deionized distilled water at a flow rate of 1.0 mL/min. The mobile phase consisted of acetonitrile and 20 μL of the filtered concentrated elute was injected by autosampler at 10 °C. The HMF calibration curve was built in the range of 0.48–750 ppm at 40 °C using a UV detector.

### 2.4. Chromaticity Testing

Color of each solution was recorded by spectrophotometry (TC-1800 MK II, Tokyo, Japan) using International Commission on Illumination (CIE) L* (lightness), a* (redness/greenness), and b* (yellowness/blueness) color scale. Both a black cup and a white tile were examined before the test to standardize the spectrophotometer. Three color measurements were measured for each sample and triplicate determinations were recorded for each sample. Total color difference (ΔE*) was calculated according to the following equation:ΔE = [(ΔL)^2^ + (Δa)^2^ + (Δb)^2^]^1/2^(1)
where ΔL = L*_sample_ − L*_control_; Δa = a*_sample_ − a*_control_; Δb = b*_sample_ − b*_control._ CIE L*_control_ a*_control_ b*_control_ was the value for the glucose–asparagine MRP solutions.

### 2.5. Statistical Analysis

Data were analyzed by analysis of variance and Duncan’s new multiple range test using SPSS 2000 statistics program for Windows, Version 12 (SPSS Inc., Chicago, IL, USA) at 95% significance level (*p* < 0.05). Linear correlations of different functional properties and the test data were determined by Pearson’s correlation at a significance level of 0.05.

## 3. Results

### 3.1. Reducing Sugar, Asparagine Content, and pH of Maillard Reaction Solutions

The reducing sugar content of asparagine, asparagine–glucose, 0.5% glucose, HMF, and HMF–asparagine solutions, which were all heated after 30 min, did not change significantly (Appendix A). Glucose (0.5%) solution heated for 30 min contained 4980–5022 μL/mL reducing sugar (Appendix A). The reducing sugar concentration of glucose (0.5%)–asparagine (0.5%) solution, which was dissolved in deionized water and heated for 30 min, decreased from 5219 μg/mL to 4785 μg/mL (*p* > 0.05) (Appendix A). Nevertheless, the reducing sugar concentration of glucose–asparagine solution (The AG group), which was dissolved in 1% acetic acid and adjusted to pH 6, decreased (from 5229 to 3263 μg/mL) by 37.6% (*p* < 0.05) (Appendix A). The reducing sugar content of chitosan–glucose–asparagine solution (The AGC group) decreased after 30 min of heating (Appendix A). A high amount of reducing sugars was also observed in the 0.5% hydroxymethylfurfural (HMF) solution and HMF–asparagine solution (Appendix A). The presence of HMF and furfural in solution indicated by magnified absorption values in 3,5-dinitrosalicylic acid (DNS) assay for reducing sugars [22]. However, the estimation of reducing sugars by DNS assay did differ significantly after 30 min of heating (Appendix A).

A significant decrease in the amount of asparagine in all of the tested groups was observed in this study, indicating the formation of MRPs produced from the reducing sugar and asparagine during 30 min of heating (Appendix A). Asparagine (0.5%) alone heated for 30 min decreased by 40.3% (from 5135 to 3064 μg/mL), which is less than glucose–asparagine solution, which decreased by 51.9% (from 5103 to 2452 μg/mL). Also, asparagine in HMF–asparagine solution decreased by 45.1% (from 4982 to 2737 μg/mL) (Appendix A). As shown in Appendix A, the hydroxymethylfurfural + asparagine and AGC: asparagine + glucose + chitosan groups showed similar patterns in the decrease of asparagine.

As shown in Appendix A, from a starting pH of 6.00 in all unheated solutions, pH decreased significantly (*p* < 0.05) in solutions containing 0.5% glucose or, HMF, but increased significantly (*p* < 0.05) in solutions containing 0.5% asparagine alone, asparagine with glucose, or HMF after 30 min of heating. 

### 3.2. Formation and Kinematic Viscosity of MRPs

Measurements of UV-absorbing intermediate compounds at OD_294_ and generation of brown-pigmented MRPs at OD_420_ showed that solutions containing HMF had extremely higher OD_294_ values than those containing asparagine–glucose–chitosan (Figure 1A,C). In solutions containing 0.5% asparagine and glucose dissolved in 1% acetic acid, absorbance values at OD_294_ and OD_420_ [14] were significantly higher than in those containing 0.5% asparagine and glucose dissolved in distilled water at any time duration after heating (Figure 1B,C and Figure 2).

Solutions of 0.5% asparagine–glucose dissolved in deionized water or in 1% acetic acid, asparagine, glucose, HMF, and HMF–asparagine displayed a similar kinematic viscosity in the range of 0.81–0.83 cSt, which differs (*p* < 0.05) from those of solutions of 0.5% chitosan and asparagine–glucose–chitosan (4.45 cSt and 4.29 cSt, respectively) (Appendix A). 

### 3.3. Effect of Chitosan on Formation of Acrylamide and HMF

Concentration of acrylamide in the solutions containing 0.5% glucose and 0.5% asparagine after 10, 20, and 30 min of heating was 28, 912, and 1459 ppb, respectively (Figure 3B). Glucose and asparagine heated with 0.5% low-molecular-weight chitosan after 10, 20, and 30 min of heating contained 33, 839, and 2370 ppb acrylamide, respectively (Figure 3B). The acrylamide content of the mixture of asparagine, glucose, and chitosan was significantly higher than that of the asparagine and glucose solution (*p* < 0.05) after 30 min of heating, which corresponds to an acrylamide formation increase of 62.4%. 

Glucose–asparagine (0.5%) dissolved in deionized water heated for 30 min generated lower acrylamide (832 ppb) when compared to the glucose–asparagine dissolved in 1% acetic acid (adjusted to pH 6) (Figure 3A,B). Nevertheless, the acrylamide concentration in aqueous HMF–asparagine solution after 30 min of heating was 103 ppb, which was lower than that of glucose–asparagine solution (*p* < 0.05) (Figure 3A). Acrylamide was not detected in the 0.5% chitosan solution alone after 30 min of heating. The addition of chitosan has the potential to enhance acrylamide formation at low concentration (0.5%) in this study.

The results revealed that HMF slowly reacted with asparagine, leading to acrylamide formation (Figure 3A). HMF was detected in the heated solutions of glucose, including asparagine–glucose, after 30 min of heating, except for 0.5% asparagine alone (Figure 4A). In solutions containing 0.5% asparagine–glucose and 0.5% glucose dissolved in distilled water at any time duration after heating, HMF content was significantly higher than that of 0.5% asparagine–glucose and 0.5% asparagine–glucose–chitosan dissolved in 1% acetic acid (Figure 4A,C). Nevertheless, HMF was detected in the 0.5% chitosan solution after 30 min of heating (Figure 4C). 

### 3.4. Chromaticity Testing of MRPs

The CIE L*a*b* values of different combination of the heated glucose, asparagine, chitosan, and HMF solutions are summarized in Appendix A. The CIE b* parameter of glucose–asparagine and glucose–asparagine–chitosan increased and the CIE L* value decreased as the heating time increase (Appendix A) as appearance of solutions shown in Figure 5. Nevertheless, the CIE a* value in Appendix A did not follow this trend. It is obvious that the CIE L* value was negatively correlated with the MRPs brown pigment (r = −0.969) and acrylamide content (Appendix A; *p* < 0.01; r = −0.923). However, CIE a* and b* and ΔE values were positively correlated to those parameters (Appendix A; *p* < 0.01). A strong negative correlation was observed between CIE L* and CIE a* (*p* < 0.01; r = -0.938), with chitosan addition in the present study (Appendix A). Nevertheless, a strong positive correlation was found between HMF and acrylamide (Appendix A; *p* < 0.01; r = 0.923). The heating time was positively related to acrylamide contents (Appendix A), but not HMF. The acryalmide content of MRPs solution was positively correlated with the heating time (Appendix A; r = 0.434; *p* < 0.05). HMF content increased upon generation of the Maillard intermediate compounds when the ingredients were dissolved in deionized water (r = 0.999) and 1% acetic acid (r = 0.960) (Appendix A). HMF content also increased upon generation of the Maillard brown pigments when the ingredients were dissolved in 1% acetic acid (r = 0.899) (Appendix A).

The heating time of a solution had a positive correlation with acrylamide content (Appendix A; *p* < 0.05; r = 0.434), absorbance values at OD_420_ (r = 0.426; *p* < 0.05), CIE b* value (r = 0.488; *p* < 0.01), ΔE (r = 0.493; *p* < 0.01) (Appendix A) but had a negative correlation with the CIE L* value (Appendix A; *p* < 0.05; r = −0.358).

## 4. Discussion

The reducing sugar content of all heated asparagine–glucose, 0.5% glucose, HMF, and HMF–asparagine solutions did not change significantly (Appendix A). Nevertheless, the reducing sugar concentration of glucose–asparagine and chitosan–glucose–asparagine solutions, which were dissolved in 1% acetic acid and adjusted to pH 6 and heated, decreased significantly (Appendix A). Kwak and Lim [23] reported the degradation of glucose by a model Maillard reaction under pH 6.5 during heating. Maillard reaction tended to be accelerated in the presence of sodium ions. It implied more glucose but not asparagine (Appendix A) dissolved in 1% acetic acid and then adjusted to pH 6 could react with asparagine to form intermediate and brown-pigmented Maillard reaction products (Figure 1C and Figure 2). A small amount of reducing sugar (51 μg/mL) was found in 0.5% chitosan solution (The C group), which increased to 274 μg/mL (*p* < 0.05) after 30 min of heating (Appendix A). This might be due to the fact that chitosan hydrolyzed into chitooligosaccharide and glucosamine and that their acetyl groups would react with dinitrosalicylic acid, thus increasing the absorbance during reducing sugar assay [24]. The increase in reducing sugar content of the 0.5% chitosan solution subjected to heat might be due to chitosan hydrolysis [25]. Yan and Evenocheck [25] reported that oligosaccharides are dominant in the hydrolysate products of chitosan hydrolysis in a weak acid at 90 °C. Chitosan could be hydrolyzed to glucosamine and acetate in a strong acid and acetyl glucosamine is found to be the incomplete hydrolysate products [25]. It also implied the formation of acrylamide at the heated glucose–asparagine dissolved in 1% acetic acid and then adjusted to pH 6 solution system was more, especially for chitosan included (Figure 3).

The pH of 0.5% HMF solution decreased significantly after 30 min of heating (Appendix A). HMF hydrolyze into formic acid and levulinic acid and release protons to decrease the pH of the solution during heating [26]. Nevertheless, the pH of solutions containing asparagine increased significantly (Appendix A). The deamidation of asparagine to aspartic acid induces pH increase when asparagine is higher than glucose in mixtures after heating at 180 °C [27]. The pH of mixtures (glucose–asparagine, and glucose–asparagine with chitosan) dissolved in 1% acetic acid and adjusted back to pH 6 decreased significantly after 20 min of heating (*p* < 0.05); nevertheless, the pH of 0.5% chitosan solution did not change significantly (Appendix A). DeMan [28] claimed that cationic amino groups (NH^3+^) would react with a carbonyl source in reducing sugars, aldehydes, or ketones, thereby making the solution more acidic. The pH of 0.5% asparagine and glucose solutions dissolved in distilled water or 1% acetic acid changed after 30 min of heating (Appendix A). In comparable trials, the products (such as acetic acid, pyruvaldehyde, and glyoxal) generated by the Maillard reaction most likely contributed to the observed decrease in solution pH [29].

The absorbance values of glucose–asparagine included 0.5% chitosan at OD_294_ and OD_420_ were significantly higher than those of glucose–asparagine solution system (Figure 1 and Figure 2). It was reported that conjugates formed with enzymatically depolymerized chitooligosaccharide demonstrated a sharp increase in UV-absorbing intermediate compounds and brown pigments [30]. Chang et al. [12] found that the addition of 1% chitosan (50–190 kDa) increased the production of intermediate compounds and brown MRP pigment in fructose with asparagine solution. Moreover, it has been demonstrated that the addition of low-molecular-weight chitosan (190 kDa) to maltose enhanced the formation of MRPs [31]. This is because shorter chitosan reacts faster than long-chain chitosan [31]. Similar results were reported with significantly higher amounts of MRPs generated in chito-oligosaccharide trials [16]. This might be due to the chitooligosaccharide, which participates more easily in the reaction than chitosans after 30 min of heating.

In the present study, absorbance values of the Maillard intermediate compounds were higher than those of the brown pigments at any time after heating (Figure 1 and Figure 2). In general, OD_420_ values of brown pigments (Figure 2: OD_420_ from 0 to 2.7) were lower than OD_294_ values (Figure 1: OD_294_ from 0 to 20.1) after 30 min of heating. In these solutions, MRPs brown pigment of 0.5% asparagine, HMF, glucose, or chitosan alone did not change during the heating duration (Figure 2). This is in consonant with our earlier findings that more Maillard reaction intermediate compounds are produced when solutions with a mixture of glucose and asparagine are heated [12,16]. The lower OD_420_ value of HMF–asparagine solution than that of glucose–asparagine mixture (Figure 2A) may indicate that less brown MRP compounds were produced after 30 min of heating (*p* < 0.05), but not after 20 min of heating. This is due to the fact that HMF, which is Maillard intermediate compound [32], can react with asparagine to form brown pigment (Figure 2A and Figure 5). Also, Qi et al. [33] demonstrated HMF was an important carbonyl intermediate of MRPs and that it reacted with asparagine to form acrylamide. Higher OD_420_ value of asparagine–glucose–chitosan solution than that of asparagine–glucose solution (Figure 2B) indicates that browner MRP compounds were produced during the final stage both after 20 and 30 min of heating. Under these heating conditions, we confirmed that the browning appearance of heated solution in the Maillard reaction is higher than caramelization of glucose (Figure 1, Figure 2, and Figure 5). However, HMF is mainly generated from caramelization of glucose than MRPs with asparagine (Figure 4). Moreover, Ajandouz et al. [19] reported that caramelization reactions accounted for 40–62% of UV-absorbance of fructose–lysine mixture heated between pH 4.0 and 7.0. In the solutions containing glucose, including asparagine–glucose and asparagine–glucose–chitosan, the Maillard intermediate compounds and brown pigments gradually increased after 30 min of heating (Figure 1). The production of colored MRPs follows the same tendency as that described by Laroque et al. [34]. Laroque et al. [34] reported that UV-absorbing intermediate compounds would increase at first and then decrease after prolonged heating. Therefore, the generation of UV-absorbing intermediate compounds might not reach a high peak at 180 °C for a heating duration of 30 min.

Chang et al. [16] demonstrated that the use of chitooligosaccharide instead of chitosans seems to significantly favor the formation of intermediate products (OD_294_) over browning pigments (OD_420_). Solutions with glucose, asparagine, and chitosan had the highest absorbance of among all the tested groups after 30 min of heating (Figure 1C). The addition of low-molecular-weight degraded chitosan increases MRPs regardless of the molecular weights of the degraded chitosan [16].

Desbrieres [35] demonstrated that low concentrations of chitosan solution exhibit a Newtonian behavior; however, increasing the chitosan concentrations up to 50 g/L led to the appearance of a non-Newtonian behavior. The kinematic viscosity of chitosan and asparagine–glucose–chitosan decreased dramatically from 4.452 to 2.200 cSt (40.3% decrease) and 4.289 cSt to 1.887 cSt (50.6% decrease), respectively, after 10 min of heating at 180 °C (Appendix A). This might be due to the fact that the chitosan was hydrolyzed into chitooligosaccharide or even glucosamine after 30 min of heating, thereby causing the kinematic viscosity of the chitosan–containing solution to decrease significantly after 10 min of heating (Appendix A). Therefore, this study confirmed viscosity of solutions containing 0.5% chitosan reduced after heating at 180 °C. The kinematic viscosity of asparagine–glucose–chitosan solution decreased to 1.473 cSt, accounting for a 65.7% decrease in viscosity. No et al. [36] demonstrated that the viscosity of 1% chitosan solution following the addition of 1% of acetic acid decreased rapidly to 91% of the initial viscosity after 15 min of autoclaving. Yan and Evenocheck [25] also found that chitosan could be fully hydrolyzed to glucosamine and acetic acid in a strong acid, to acetyl glucosamine when the hydrolysis is incomplete, and to chitooligosaccharide in a weak acid and at a low temperature. The kinematic viscosity of glucose–asparagine–chitosan seemed to decrease more dramatically than that of 0.5% chitosan alone. Such difference might be due to the decreased pH of the Maillard reaction (Appendix A). The pH of a heated glucose–asparagine–chitosan solution is significantly lower than that of a heated 0.5% chitosan solution (*p* < 0.05).

Both 1% chitooligosaccharide and 1% low molecular chitosan were shown to mitigate the formation of acrylamide in 1% glucose and 1% asparagine solution [12,16]; in addition, Mogol and Gökmen [11] concluded that chitosan addition did not significantly affect the acrylamide formations in biscuits and crust models. However, glucose and asparagine react with hydrolyzed chitosan to form more acrylamide in this study (Figure 3). This is the first time to observe that low-molecular-weight chitosan enhances the formation of acrylamide after 30 min of heating at 180 °C. Incorporation of chitosan after 30 min of heating results in products with a higher amount of acrylamide, but not reducing sugar or HMF. It has been reported that the rate of acid hydrolysis of chitosan was affected by the concentration of chitosan [37]. This might support the 0.5% chitosan hydrolyzed into lower molecular weight chitosan, chitooligosaccharide, or glucosamine in the solution after 30 min of heating to react with asparagine and generate more acrylamide (Figure 3B), which was consistent with our data in Figure 1 and Figure 2. Therefore, addition of 0.5% chitosan appears to not be suitable to mitigate acrylamide formation in the food model system.

Addition of sodium ion prior to thermal processing increased the formation of acrylamide and HMF in a glucose–asparagine model system [38]. This is similar to the 0.5% glucose and 0.5% asparagine dissolved in 1% acetic acid and then adjusted the solution pH to 6 by adding sodium hydroxide solution in this study (Figure 3 and Figure 4). It is similar to adding sodium ion prior to thermal processing and it enhances the Maillard reaction. Gökmen et al. [32] reported that the activation energy of acrylamide formation was 138.78 and 83.94 kJ/mol for the model systems of HMF–asparagine and glucose–asparagine, respectively. It is confirmed that HMF can react with asparagine to form acrylamide, which is less than the formation pathway of glucose–asparagine solution system (Figure 3). The acrylamide formation follows the Arrhenius law, with very high correlation coefficients from 90–180 °C [32] and the HMF concentration in the glucose–asparagine model system showed an increased trend, followed by a decrease trend, which was different from the gradual accumulation of acrylamide [33]. This trend showed that HMF was an intermediate compound of MRPs and that it is subject to further reactions after formation. This might support observation that the reaction of HMF with asparagine is not the main route for acrylamide formation and the HMF is intermediate MRPs.

In this study, all the test solutions with 0.5% glucose, asparagine, and chitosan developed much lower acrylamide, MRPs, and HMF content after 30 min of heating than previous studies with 1% mixtures [12,16]. Therefore, the concentration of asparagine and glucose dominates the formation of acrylamide. Acrylamide therefore seems to be generated primarily from the reaction of glucose with asparagine [38,39]; however, glucosamine hydrolyzed from chitosan after 30 min of heating may also contribute to the generation of acrylamide. Although Yasuhara et al. [40] reported that heating asparagine powder alone could produce acrylamide via thermal degradation, in this study, acrylamide was not detected in the heating of 0.5% chitosan alone and 0.5% asparagine solution alone after 30 min (Figure 3A,B). Acrylamide levels significantly increased when glucosamine (124%) were added to California-style black ripe olives prior to sterilization [39]. Glucosamine and N-acetyl glucosamine are the main amino sugars present in olives. These compounds contain nitrogen and carbon atoms and have reactive functional groups, such as amine, hydroxyl, and carbonyl, which could participate in the formation of acrylamide in California-style black ripe olives through inter- and intra-chemical reactions [39], which might support the result high acrylamide content was found in heated glucose–asparagine–chitosan solution system. 

The formation of HMF from glucose alone appears to be much more than from glucose–asparagine (Figure 4A,C) and followed different pathways in these model systems. The carbonyl group of the reducing sugar glucose would react with the amino group of chitosan, chitooligosaccharide, and glucosamine to form HMF and chitosan–glucose, or chitooligosaccharide-glucose conjugates of MRPs and the conjugates would also generate HMF during the heating process. However, the formation of HMF from different formation pathways by all of these conjugates of MRPs is significantly lower than the HMF formation pathway from caramelization of glucose.

HMF can react with asparagine to form acrylamide as confirmed by Gökmen et al. [32]. However, the asparagine content of the model of HMF–asparagine was depleted, which is similar to that of asparagine–glucose (Appendix A; *p* > 0.05). Moreover, heating asparagine could produce acrylamide via thermal degradation [40].

It was demonstrated that equimolar asparagine–HMF model system generated acrylamide more efficiently than the asparagine–glucose model system during heating at 180 °C under low moisture condition [32]. However, the HMF and other carbonyls from sugar dehydration could not be considered as potent contributors to acrylamide formation in thermally processed foods in this study (Figure 3A).

HMF can react with asparagine to form acrylamide (Figure 3A) and this was also reported by Gökmen and Senyuva [38]. However, it was not shown that the addition of chitosan can significantly induce the formation of HMF (Figure 4C). HMF levels in the 0.5% hydroxymethylfurfural alone or with asparagine did not change significantly after heating (Figure 4B). HMF might react with asparagine to form MRPs (Figure 1 and Figure 2) and acrylamide (Figure 3A). HMF was mainly formed via caramelization of glucose in this study (30 ppm) (Figure 4A) after 30 min of heating compared to that HMF generates through the formation of a dicarbonyl intermediate, 3-deoxyglucosone, from Maillard reaction and caramelization (7 ppm). Gentry and Roberts [41] reported that there was a negative correlation between HMF formation rates and asparagine concentration, which suggests a difference in the chemical pathway. This may be due to the fact that the pH of asparagine–glucose solution was 6.55 after 30 min of heating, which was higher than that (pH 4.41) of 0.5% glucose alone after 30 min of heating (Appendix A). The heated 0.5% glucose after 30 min was shown to generate more HMF and lower pH values than the solution of 0.5% asparagine and glucose and this might be due to the fact the HMF tends to be formed at an acidic pH.

A higher amount of HMF was formed at low pH in a model system [26]. HMF formation has been proposed to result from caramelization of reducing sugars [1] and this was also confirmed in this study (Figure 4A). HMF is reactive and the concentration of precursors (0.5% asparagine and 0.5% glucose) is high, such that so it could convert sugars into HMF at similar rate as its elimination when chitosan was added, although chitosan might hydrolyze into lower molecular weight chitosan (Figure 4C). The carbonyl group of reducing sugars, ketones, or aldehydes reacts with the amino group of chitosan to form HMF and chitosan–sugar conjugates of MRPs [42], a mechanism different from that generating acrylamide in MRPs produced from glucose and asparagine. However, the carbonyl group of reducing sugars ketones or aldehydes and glucosamines might react with the amino group of glucosamines to form acrylamide and HMF. HMF is an intermediate, which could also react its amines. Figure 4C shows that the hydrolyzed chitosan reacts to form HMF, especially after 30 min of heating. Chitosan may decompose to release a free glucosamine as a reactive intermediate. Glucose in the presence of water can be converted into fructose [42] and can be protonated to a fructofuranosyl cation, which reacts with asparagine to form fructofuranosyl amine, which could rearrange into a Heyns product prior to the formation of acrylamide [42]. As a result, the fructofuranosyl cation reacted with chitosan and chitooligosaccharide in the current study. HMF was detected in the heated model solution of asparagine and glucose.

Whether this is due to the lower molecular weights of the chitosan is yet to be confirmed. However, the significantly higher amount of acrylamide and MRPs and the lower viscosity of solutions generated in the chitosan trials might be due to the formation of oligochitosan or glucosamine and sodium ion addition for pH adjustment, which is much easier to participate in Maillard reaction than chitosan. The content of HMF in the MRPs (ppm level) was significantly higher than that of acrylamide (ppb level) (Figure 3 and Figure 4) mainly from different pathways of Maillard reaction. 

## 5. Conclusions

In our proposed model system, 0.5% (*w*/*v*) glucose, asparagine, chitosan, and hydroxymethyl furfural (HMF) were used to evaluate the potential interactions between these molecules in different heating time and experimental settings. This research was designed to understand the formation of HMF and acrylamide and the functional properties of MRPs by simulating the liquid food thermal processing parameters. Low molecular chitosan contains amino groups that can react with the carbonyl groups of glucose to generate more MRPs. Following this concept, the presence of HMF in solutions was observed during 30 min of heating; however, the amount of HMF did not change significantly, but was generated dramatically during the first 20 min of heating without the addition of HMF and even the chitosan alone after 20 min of heating. As a result, the kinematic viscosity of chitosan-containing solution decreased significantly after 10 min of heating and it caused the formation of HMF and acrylamide dramatically after 20 min of heating. For potential food processing applications, the formation of undesirable acrylamide over 1000 ppb was after 20 min of heating and the formation of HMF was way below 50 ppm during 30 min of heating. More importantly, the solutions containing asparagine and reducing sugar suggested that heating for over 20 min is not a good strategy for preparing healthier food.

## Figures and Tables

**Figure 1 polymers-13-01901-f001:**
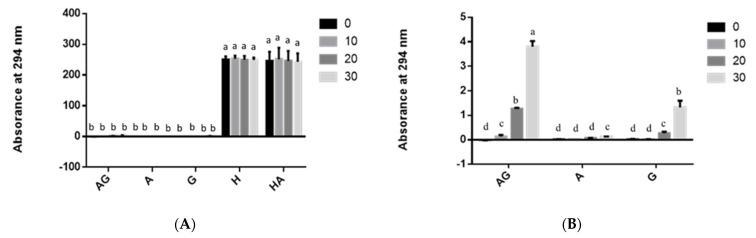
The UV absorbance at 294 nm of different groups in: (**A**) deionized water, (**B**) deionized water not including hydromethylfurfural groups, and (**C**) acetic acid at various time points during heating processing. AG: Asparagine + Glucose; A: asparagine; G: glucose; H: hydroxymethylfurfural; HA: hydroxymethylfurfural + asparagine; C: chitosan; AGC: asparagine + glucose + chitosan. Values are expressed as mean ± standard deviation (SD) (*n* = 3). ^a–e^ Indicate significant differences between different groups (*p* < 0.05).

**Figure 2 polymers-13-01901-f002:**
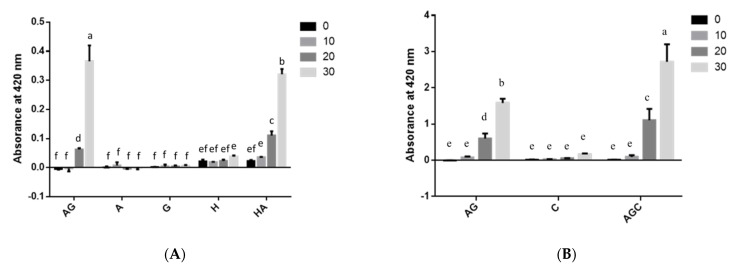
The absorbance at 420 nm of different groups in: (**A**) deionized water and (**B**) acetic acid at various heated processing duration. AG: asparagine + glucose; A: asparagine; G: glucose; H: hydroxymethylfurfural; HA: hydroxymethylfurfural + asparagine; C: chitosan; AGC: asparagine + glucose + chitosan. Values are expressed as mean ± standard deviation (SD) (*n* = 3). ^a–f^ Indicate significant differences between different groups (*p* < 0.05).

**Figure 3 polymers-13-01901-f003:**
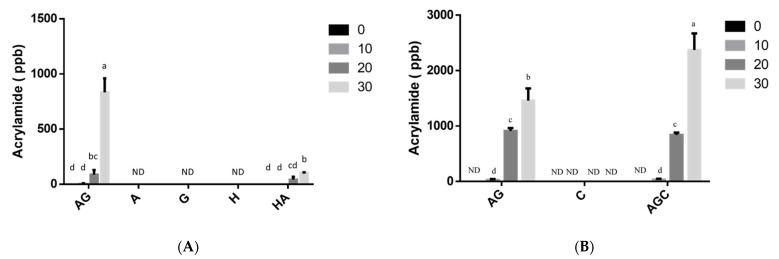
Amounts of acrylamide (ppb) between different groups in: (**A**) deionized water and (**B**) acetic acid at various heated processing duration. AG: asparagine + glucose; A: asparagine; G: glucose; H: hydroxymethylfurfural; HA: hydroxymethylfurfural + asparagine; C: chitosan; AGC: asparagine + glucose + chitosan. ND: Not Detected. Values are expressed as mean ± standard deviation (SD) (*n* = 3). ^a–d^ Indicate significant differences between different groups (*p* < 0.05).

**Figure 4 polymers-13-01901-f004:**
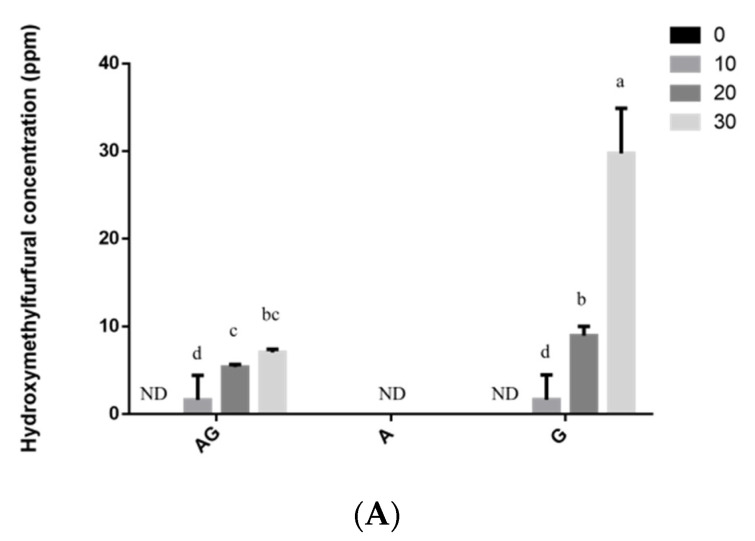
Amounts of hydroxymethylfurfural (ppm) between different groups in: (**A**) AG, G, and A groups dissolved in deionized water, (**B**) HA and H groups dissolved in deionized water, and (**C**) dissolved in acetic acid heated processing duration. AG: asparagine + glucose; A: asparagine; G: glucose; H: hydroxymethylfurfural; HA: hydroxymethylfurfural + ssparagine; C: chitosan; AGC: asparagine + glucose + chitosan. Values are expressed as mean ± standard deviation (SD) (*n* = 3). ^a–d^ Indicate significant differences between different groups (*p* < 0.05).

**Figure 5 polymers-13-01901-f005:**
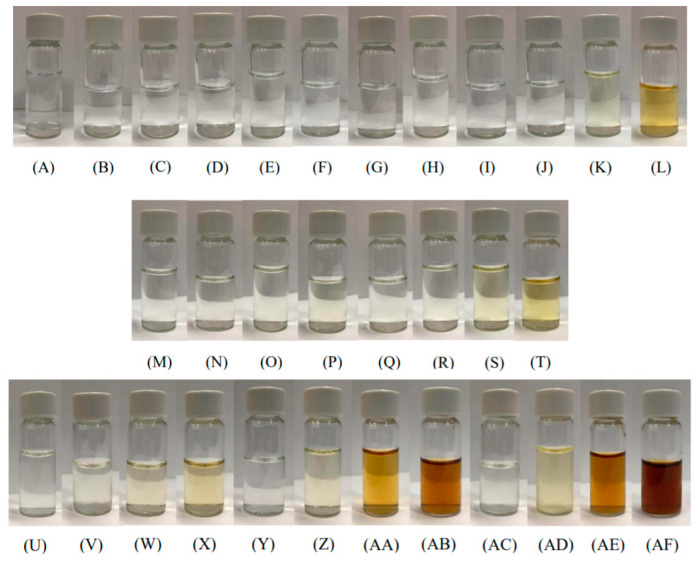
Appearance of solution groups heated at different processing duration. V: acetic acid; AG: asparagine + glucose; A: asparagine; G: glucose; H: hydroxymethylfurfural; HA: hydroxymethylfurfural + asparagine; C: chitosan; AGC: asparagine + glucose + chitosan. (**A**) G 0 min; (**B**) G 10 min; (**C**) G 20 min; (**D**) G 30 min; (**E**) A 0 min; (**F**) A 10 min; (**G**) A 20 min; (**H**) A 30 min; (**I**) AG 0 min; (**J**) AG 10 min; (**K**) AG 20 min; (**L**) AG 30 min; (**M**) H 0 min; (**N**) H 10 min; (**O**) H 20 min; (**P**) H 30 min; (**Q**) HA 0 min; (**R**) HA 10 min; (**S**) HA 20 min; (**T**) HA 30 min; (**U**) C 0 min; (**V**) C 10 min; (**W**) C 20 min; (**X**) C 30 min; (**Y**) AGV 0 min; (**Z**) AGV 10 min; (**AA**) AGV 20 min; (**AB**) AGV 30 min; (**AC**) AGC 0 min; (**AD**) AGC 10 min; (**AE**) AGC 20 min; (**AF**) AGC 30 min.

## Data Availability

The data present in this work are available on request from the corresponding author.

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
