# Peer review of "Effect of Hydroxymethylfurfural and Low-Molecular-Weight Chitosan on Formation of Acrylamide and Hydroxymethylfurfural during Maillard Reaction in Glucose and Asparagine Model Systems"

_polymers, 2021, doi:10.3390/polym13121901_

Round 1

Reviewer 1 Report

1) The general implications of the results should be emphasized. In its current form the manuscript reads more as a report than a scientific article.

2) The figures are of low quality and should be reorganized according to publication standards. More care should be taken in constructing the figures.

3) The Discussion section focuses more on the literature than the actual results. The literature should be introduced under the Introduction section, and the Discussion section should focus on the description of the actual results obtained by the authors for the first time. The novelty of the work (novel findings) are unclear in the current form of the manuscript.

4) What is the effect of the base, i.e. sodium hydroxide, on the system? How are the results affected by the use of different bases? What was the rationale for the selection of the sodium hydroxide? The authors should give more information on these aspects.

5) A similar study investigating the effect of chitosan was reported and should be mentioned (10.1039/c6fo00755d).

6) The authors should at least mention with an example that HMF is also a biomass-derived building block and not only a carcinogenic impurity (10.1002/cssc.202000453).

7) Reproducibility of the results are demonstrated, and error bars are provided. However, are those standard deviations? Were independently prepared materials or repeated measurements on the same batches of materials were used? These should be explicitly mentioned in the corresponding figure captions.

8) The first paragraph of section 2.3 starts with irrelevant general information such as ‘The Materials and Methods should be described with sufficient details to allow 104 others to replicate and build on the published results.’ Which should be all deleted.

Author Response

Dear Reviewer #1 The authors are extremely grateful to anonymous referee involved for providing his/her excellent comments and valuable advice in this paper. We have revised the paper based on the referee’s comments. We have pleasure in requesting the referee to review this paper. Thank you. Your prompt attention to this paper will be much appreciated. Point 1: The general implications of the results should be emphasized. In its current form the manuscript reads more as a report than a scientific article. Response 1: We have rewritten and emphasized the general implications of the results through the Introduction, Results and Discussion sections in our article carefully as revised texts marked in yellow color in the revised manuscript. Thanks for the suggestions and we appreciate you giving us the second chance to revising the manuscript. (Please see the revised manuscript). Point 2: The figures are of low quality and should be reorganized according to publication standards. More care should be taken in constructing the figures. Response 2: All Figures, supplementary Figures and Tables were reorganized and adjusted to give an aesthetic and clear presentation in the revised manuscript. Thanks for informing the problems in the Figures and Tables in the previous manuscript. Point 3: The Discussion section focuses more on the literature than the actual results. The literature should be introduced under the Introduction section, and the Discussion section should focus on the description of the actual results obtained by the authors for the first time. The novelty of the work (novel findings) are unclear in the current form of the manuscript. Response 3: Thank you for pointing out the comments related to discussion section. We have enhanced the novelty and significance of this work by adding more description and discussion to highlight the major findings in this study. Hopefully, the Discussion section has been much improved (Please see the yellow color texts of the Introduction section lines 77-80 at page 2 and Discussion section from pages 10 to 14 of revised manuscript). Point 4: What is the effect of the base, i.e. sodium hydroxide, on the system? How are the results affected by the use of different bases? What was the rationale for the selection of the sodium hydroxide? The authors should give more information on these aspects. Response 4: We have rewritten and deleted the misleading words by adding different concentrations of sodium hydroxide (1N and 0.001N) to adjust the solution pH to 6.0 at Materials and Methods section at page 3 lines 103-105 of the revised manuscript. The reason for using different concentrations of sodium hydroxide is to adjust pH back to 6 quickly. However, this processing will increase sodium ion concentration in the solution. We are so sorry for the mistakes. (Please see the revised manuscript). Point 5: A similar study investigating the effect of chitosan was reported and should be mentioned (10.1039/c6fo00755d). Response 5: The paper (10.1039/c6fo00755d) has been cited at page 2 lines 54 and page 12 lines 458-461 of the revised manuscript in the second paragraph of Introduction section and at page 12 Discussion section as the marked yellow texts. Thanks for informing us the important related study. Point 6: The authors should at least mention with an example that HMF is also a biomass-derived building block and not only a carcinogenic impurity (10.1002/cssc.202000453). Response 6: We have added and cited the biomass-derived building block information of HMF at page 2 lines 47 and 48 of the revised manuscript in the last sentence in the first paragraph of Introduction section as the marked yellow texts. Thanks for the suggestions and we appreciate you giving us the information except its harmful effects. Point 7: Reproducibility of the results is demonstrated, and error bars are provided. However, are those standard deviations? Were independently prepared materials or repeated measurements on the same batches of materials were used? These should be explicitly mentioned in the corresponding figure captions. Response 7: The error bars of sample replicated information in Figures and supplementary Figures and Tables for the data collection were added in caption section as yellow color texts. All determinations were replicated three times, mean values and standard deviations were reported in supplementary Tables 1& 2. Thanks for informing the problems in the Figures 1 to 4 and Tables of the previous manuscript. Point 8: The first paragraph of section 2.3 starts with irrelevant general information such as ‘The Materials and Methods should be described with sufficient details to allow others to replicate and build on the published results.’ Which should be all deleted. Response 8: We are sorry for forgetting to remove the explanation of Materials and Methods section. The irrelevant general information of the first few sentences in section 2.3 has been deleted. Thanks for informing us serious mistake in the previous manuscript. (Please see the revised manuscript at page 3 lines 109 to 110). Yours truly, Wen-Chieh Sung, Ph.D. Professor Department of Food Science National Taiwan Ocean University

Reviewer 2 Report

  1. The novelty and meaning of the manuscript is not presented well. The introduction part should be revised accordingly to highlight its novelty.
  2. The title of this manuscript is “Effect of the addition of hydroxymethylfurfural and low-molecular-weight chitosan on formation of acrylamide and hydroxymethylfurfural during Maillard reaction in glucose and asparagine model systems. It was described that Acrylamide and 5-hydroxymethylfurfural (HMF) are undesirable carcinogens. If so, what is the point of addition of hydroxymethylfurfural in the system? It is a little confused and the authors should add more information in the introduction part to give a clear description.

  1. Why choose low-molecular-weight chitosan (50–190 kDa) rather than higher -molecular-weight chitosan?
  2. As the introduction described: The browning intensity of Maillard reaction products can be increased by low-molecular-weight chitosans (50–190 kDa), which adversely reduce the formation of acrylamide in 1% glu-59 cose-asparagine and fructose-asparagine food model systems.

It was also described that Acrylamide and 5-hydroxymethylfurfural (HMF) are undesirable carcinogens.

If so, it seems that reducing the formation of acrylamide is good and desirable. So the word “ adversely” used here is confused.

  1. All the Figures and pictures should be revised to give a clear and aesthetic presentation. For example, the author should adjust the font size of horizontal and vertical axes.
  2. There are too much data in table 1 and 2. If possible, it is suggested to present the results in figures rather than tables to give a better presentation.

Author Response

Dear Reviewer #2 The authors are extremely grateful to anonymous referee involved for providing his/her excellent comments and valuable advice in this paper. We have revised the paper based on the referee’s comments. We have pleasure in requesting the referee to review this paper. Thank you. Your prompt attention to this paper will be much appreciated. Point 1: The novelty and meaning of the manuscript is not presented well. The introduction part should be revised accordingly to highlight its novelty. Response 1: Thanks for the valuable suggestions. We have rewritten and elucidated the novelty attempt of this study through the second paragraph of introduction section in our article carefully as added texts (lines 77 to 80) marked as yellow color in the revised manuscript. The novelty and meaning of the manuscript is also added in Discussion section as yellow color texts accordingly to highlight of the revised manuscript. Thanks for the suggestions and we appreciate you giving us a chance to resubmitting the manuscript. (Please see the revised manuscript at page 2). Point 2: The title of this manuscript is “Effect of the addition of hydroxymethylfurfural and low-molecular-weight chitosan on formation of acrylamide and hydroxymethylfurfural during Maillard reaction in glucose and asparagine model systems. It was described that Acrylamide and 5-hydroxymethylfurfural (HMF) are undesirable carcinogens. If so, what is the point of addition of hydroxymethylfurfural in the system? It is a little confused and the authors should add more information in the introduction part to give a clear description. Response 2: Thanks for the value comments again. This study aims to investigate the possible reaction routes for the formation and HMF and acrylamide by addition of HMF and LMW chitosan and monitor the levels of HMF and acrylamide. In this study, the formation of HMF from caramelization of glucose or from glucose-asparagine can be distinguished in Figure 4. And the content of acrylamide generated from HMF-asparagine and glucose-asparagine model systems was also investigated to see the role of HMF in Maillard reaction during heating. Thanks for informing the problems and suggesting us to give a clear description in the Introduction section. We added the previous information at the page 2 (lines 77 to 80) of introduction section of the revised manuscript. Also, we modified the title for clarification. Point 3: Why choose low-molecular-weight chitosan (50–190 kDa) rather than higher -molecular-weight chitosan? Response 3: Thank you for pointing out the question. We have found the lower molecular weight chitosan can mitigate the formation of acrylamide in 1% glucose-asparagine and fructose-asparagine food model systems (Chang et al., 2016; Sung et al., 2018) better than the higher molecular weight of chitosans (190–310 kDa, and 310–375 kDa). In this study, we further investigated the possible effects of lower concentration of the LMW chitosan on acrylamide on HMF and acrylamide formation. Point 4: As the introduction described: The browning intensity of Maillard reaction products can be increased by low-molecular-weight chitosans (50–190 kDa), which adversely reduce the formation of acrylamide in 1% glucose-asparagine and fructose-asparagine food model systems. It was also described that Acrylamide and 5-hydroxymethylfurfural (HMF) are undesirable carcinogens. If so, it seems that reducing the formation of acrylamide is good and desirable. So the word “ adversely” used here is confused. Response 4: We can feel the referee spent so much time reading our manuscript very carefully. We have rewritten and deleted the misleading word “adversely” and added “simultaneously” at page 2 line 62 in the introduction section of the revised manuscript. We are sorry for the mistakes. Hopefully, the revised work is much appropriate description. Thanks for the suggestions and we appreciate you giving us the great comment. (Please see the second paragraph of page 2 in the revised manuscript). Point 5: All the Figures and pictures should be revised to give a clear and aesthetic presentation. For example, the author should adjust the font size of horizontal and vertical axes. Response 5: All figures were reorganized and adjusted the font size to appropriate size. Thanks for informing us the problems in the previous version of manuscript figures. Please see the Figures from 1 to 5 of the revised manuscript. Point 6: There are too much data in table 1 and 2. If possible, it is suggested to present the results in figures rather than tables to give a better presentation. Response 6: We have removed Tables 1 & 2 to supplementary Tables and added Figure 5 in the revised manuscript. Hopefully, the pictures of Figure 5 will be clear in the current form instead of too much numbers. Thanks for the suggestions and we appreciate you for all the great comments. (Please see the revised manuscript). Yours truly, Wen-Chieh Sung, Ph.D. Professor Department of Food Science National Taiwan Ocean University

Round 2

Reviewer 1 Report

The comments have been addressed. Proofreading is still necessary.